

# Understanding the role of exosomal lncRNAs in rheumatic diseases: a review

Ruofei Chen[1,*], Dongqing Zhou[1,*], Yangfan Chen[1], Mingwei Chen[2] and Zongwen Shuai[1]

[1] Department of Rheumatology and Immunology, The First Affiliated Hospital of Anhui Medical University, Hefei City, Anhui Province, China
[2] Department of Endocrinology, The First Affiliated Hospital of Anhui Medical University, Hefei City, Anhui Province, China
* These authors contributed equally to this work.

Corresponding authors
Mingwei Chen, chmw1@163.com
Zongwen Shuai,
shuaizongwen@ahmu.edu.cn

## ABSTRACT

Rheumatic diseases, a group of diseases whose etiology is still unclear, are thought to be related to genetic and environmental factors, leading to complex pathogenesis. Based on their multi-system involvement, the diagnosis and treatment continue to face huge challenges. Whole-genome assays provide a distinct direction for understanding the underlying mechanisms of such diseases. Exosomes, nano-sized bilayer membrane vesicles secreted by cells, are mentioned as a key element in the physiological and pathological processes of the body. These exosomes mediate biologically active substances, such as nucleic acids, proteins, and lipids and deliver them to cells. Notably, long non-coding RNAs (lncRNAs), a unique class of non-coding RNAs, have been implicated in the pathogenesis of rheumatic diseases. However, the mechanism needs to be further explored. This article provided a comprehensive review of the findings on exosomal lncRNAs in rheumatic diseases, including rheumatoid arthritis, osteoarthritis, systemic lupus erythematosus, autoimmune liver diseases, primary dermatomyositis, and systemic sclerosis. Through in-depth understanding of these lncRNAs and their involved signaling pathways provide new theoretical supports for the diagnosis and treatment of rheumatic diseases.

## INTRODUCTION

### Exosome and lncRNAs

Exosomes, a subtype of extracellular vesicles, are formed through the double invagination of the plasma membrane and the subsequent synthesis of intracellular multivesicular bodies containing intraluminal vesicles (*Kalluri & LeBleu, 2020*). Upon fusion of these vesicles with the cell membrane, exosomes with a diameter of 40–160 nm are secreted through exocytosis. Exosomes are extensively distributed *in vivo* and can be secreted by almost all cell types and body fluids (*Zhang et al., 2020*).

Exosome, being excellent carriers, exhibit low immunogenicity, and have a wide distribution. They can be stabilized *in vivo* by the presence of surface-specific substances, CD55 and CD59, for instance, enable exosomes to demonstrate superior performance

**Table 1 The traditional isolation method of exosome.**

| Method | Advantage | Disadvantage | Feature | References |
|---|---|---|---|---|
| Ultracentrifugation | Widely used, Avoiding cross contamination | High time consumption, High cost, Structural damage, | Suitable for separations with significant differences in sedimentation coefficients | *Zhang et al. (2020)*, *Livshits et al. (2016)* |
| Density gradient centrifugation | High purity | High cost | Using in combination with ultracentrifugation to improve the purity | *Zhang et al. (2020)*, *Livshits et al. (2016)* |
| Polymer precipitation | Low cost | Low purity | Using reagents to reduce solubility in the centrifuged state | *Zhang et al. (2020)*, *Rider, Hurwitz & Meckes (2016)* |
| Size-exclusion chromatography | Complete tructure, simple operation, fast speed | Low purity | Using different molecular sizes to screen small molecules into the gel pores for elution separation | *Zhang et al. (2020)*, *Böing et al. (2014)* |
| Ultrafiltration | Low cost, High enrichment efficiency, Complete activity | Low purity | Screening with ultrafiltration membranes | *Zhang et al. (2020)*, *EV-TRACK Consortium et al. (2017)* |
| Immunoaffinity chromatography | Strong specificity, High sensitivity, High purity, High yield, Low sample requirements | Harsh storage condition | Antigen-antibody specific binding to achieve separation | *Zhang et al. (2020)*, *Li et al. (2017)* |

during blood transport as nanobodies (*Clayton et al., 2003*). This unique feature allows exosomes to escape capture by the mononuclear macrophage system and avoid activation by the complement system. In addition, exosomes possess remarkable ability in penetrating physical and biological barriers, facilitating their widespread distribution and supporting their incorporation in artificial designs aimed at transmitting information between cells.

Traditional exosome extraction methods have their advantages and disadvantages, which have been summarized in Table 1 (*Zhang et al., 2020*; *Livshits et al., 2016*; *Rider, Hurwitz & Meckes, 2016*; *Böing et al., 2014*; *EV-TRACK Consortium et al., 2017*; *Li et al., 2017*). However, none of these methods can be efficiently and in batches clinically, and too many impurities in the extraction process cannot be used for biomarker detection. In order to narrow the gap with the real world, some researches try to explore new extraction methods. A TIM4 affinity separation method for phosphatidylserine (PS), a component of exosome membrane, was proposed (*Yoshida & Hanayama, 2022*). When TIM4 binds to PS in a Ca2+-dependent manner, complete exosomes can be eluded from TIM4 beads by adding the chelating agent ethylenediamine tetraacetic acid (EDTA). It has the advantages of simple operation, low cost and high purity, and can be used in clinical large-scale detection of biomarkers.

Exosomes play a crucial role in intercellular RNA transfer. Among these RNAs, long non-coding RNAs (lncRNAs) are a class of non-coding RNAs (ncRNAs) with a length of more than 200 nucleotides, highly conserved sequences and are widely present in the

nucleus or cytoplasm. With the advancements in high-throughput sequencing, mass spectrometry, and bioinformatics, numerous lncRNAs have been discovered to have significant involvement in the epigenetic regulation of transcription, post-transcription, translation, post-translation, and epigenetic modification. These lncRNAs are widely involved in biological processes under physiological and pathological regulation *in vivo*, such as cell proliferation, apoptosis, differentiation, autophagy, development, and aging (*Engreitz et al., 2016*; *Liu, Qian & Cao, 2016*; *Gao et al., 2018*). Besides, in recent years, extensive studies have been conducted on exosomal lncRNAs, revealing their involvement in abnormal replication of tumor cells and the expression of genes. Owing to the remarkable stability of exosomes, the potential of utilizing exosome-mediated transport of special lncRNAs are proposed as future therapeutic targets.

## Rheumatic diseases and exosomal lncRNAs

Similar to tumors, rheumatic diseases exhibits complex pathogenesis, and their unclear early diagnosis and ineffective treatment effect, even the drugs treatment for rheumatic diseases were originally derived from tumor therapy. Many studies have been conducted to determine the role of exosomes in rheumatic diseases.

Exosomes are proved that involving in the pathogenesis of rheumatoid arthritis (RA) through intracellular lncRNAs, such as the NF-κB and Wnt signaling pathways (*Ren et al., 2023*; *Hou et al., 2020*), thus affecting the expression imbalance of Th17/Treg cells and leading to the damage of synovial cells. In addition to mechanistic studies, the exosomes derived from normal stem cells have demonstrated therapeutic potential in osteoarthritis, and they can be absorbed by the chondrocytes of osteoarthritis model mice model, stimulate cell migration and proliferation and reduce chondrocyte apoptosis (*Zhu et al., 2017*). Owing to the excellent performance of lncRNAs in regulating gene expression, many studies have been devoted to investigating whether lncRNAs are involved in the physiological and pathological processes of rheumatic diseases. In systemic lupus erythematosus, the lncRNA TUG1 acts as a protective gene by inhibiting the NF-κB pathway, thereby delaying the progression of lupus nephritis (*Cao et al., 2020*). TMEVPG1 is positively correlated with disease activity marker in patients with Sjögren's syndrome, including the number of CD4+ T cells, anti-SSA antibodies, erythrocyte sedimentation rate (ESR), total IgG amount, and Th1 cell ratio (*Wang et al., 2016*).

Due to the wide range of exosome sources and the fact that lncRNAs can participate in multiple pathways, the role of exosomal lncRNAs *in vivo* is complicated. LncRNA NEAT1 carried by exosomes derived from human umbilical vein endothelial cells can induce M2-type polarization of macrophages and alleviate joint inflammation (*Chen et al., 2023*). Another study on the exosome LncRNA NEAT1 showed that it regulates the intestinal epithelial barrier and macrophage polarization promotes the inflammatory response in inflammatory bowel disease (*Liu et al., 2018b*). In addition, in rheumatoid arthritis, the exosome LncRNA NEAT1 can also promote the proliferation of CD4+ T cells and the differentiation of Th17 cells through the WNT signaling pathway, and aggravate the occurrence of bone destruction (*Liu et al., 2021*). In conclusion, the same exosome lncRNA can affect different cells by acting on different pathways, but it can lead to a variety of

rheumatic diseases. In order to further determine the role of a specific cell, some studies have resorted to special materials such as hydrogels to achieve targeted therapy (*Zhu et al., 2022*). But these studies have not been confirmed on a large scale and have not yet been used clinically.

Therefore, both exosomes and lncRNAs are closely related to the future development of rheumatic diseases. In this article, we summarized the recently discovered research results on the relationship between exosomal lncRNAs and rheumatoid arthritis, osteoarthritis, autoimmune liver disease, primary dermatomyositis, and systemic sclerosis in Table 2, and plotted the pathways involved in these lncRNAs in different cells and the processes affecting micRNAs and target genes in Fig. 1. We hope this article will help to explore the complex pathogenesis of related diseases and find new diagnostic markers and therapeutic targets.

## SURVEY METHODOLOGY

We searched PubMed, the Web of Science, Embase, and Cochrane by the searching the following: "exosome", "lncRNA", "rheumatoid arthritis/osteoarthritis/systemic lupus erythematosus/primary biliary cholangitis/dermatomyositis/systemic sclerosis" are searched in combination with the subject title or its free term respectively (the specific searching strategies can be found in the supplemental searching strategies).

### RA and exosomal lncRNA

RA is a general autoimmune disease that affects multiple systems and is common in middle-aged and elderly female patients. It is characterized by synovitis and irreversible joint damage and mainly manifests as redness, swelling, heat, and pain in the facet joints with stiffness in the morning and with or without multisystem damage (*Wu et al., 2022b*). This joint injury is mainly caused by the interaction of T cells, B cells, neutrophils, fibroblast synoviocytes and osteoclasts, producing antibodies to attack themselves and activating the immune pathway (*van Delft & Huizinga, 2020*). Since this damage to oneself can persist without interference and the body will not heal itself, diagnosis and treatment are particularly important for rheumatoid arthritis. In patients with RA, the synovial tissues are characterized by the presence of fibroblast-like synovial cells (FLSs), which produce excessive amounts of cytokines, leading to persistent inflammation and irreversible damage to the joint structure (*Jay, Britt & Cha, 2000*). Researchers aimed to restore the homeostasis of the affected joint synovium by targeting such cells. Similarly, in the related studies of exosomal lncRNA, FLSs are often selected as research subjects, hoping to explore the possibility of diagnosis and treatment in RA by finding special lncRNA derived from or acting on FLSs.

Exosomal lncRNA TRAFD1-4:1 derived from FLSs can degrade the chondrocyte matrix, affect the migration of chondrocytes, and cause irreversible damage to the joints (*Ren et al., 2023*). CXCL-1 promotes the development of joint inflammation in RA by releasing inflammatory factors such as IL-6 and IL-17 (*Hou et al., 2020*; *Kuwabara et al., 2017*). In *Ren et al. (2023)*, TRAFD1-4:1 was found as a highly expressed non-coding gene in RA, which affected the expression of the target gene CXCL-1 by inhibiting miR-27a-3p,

**Table 2  Current studies of exosomal lncRNAs in rheumatic diseases.**

| Disease | Official symbol | Origin | Effection | Expression | Related gene or signaling pathways | References |
|---|---|---|---|---|---|---|
| RA | TRAFD1-4:1 | FLS | Chondrocyte | Up-regulated | miR-27a-3p/CXCL1 | *Ren et al. (2023)* |
| RA | NEAT1 | PBMC | FLS | Up-regulated | miR-23a/MDM2/SIRT6/NF-κB signaling pathway | *Rao et al. (2020)* |
| RA | NEAT1 | Serum | Tcell | Up-regulated | miR-144-3p/ROCK2/Wnt-β-catenin signaling pathway | *Liu et al. (2021)* |
| RA | Hotair | Plasma | Synoviocyte, osteoclast | Up-regulated | MMP-2/MMP-13 | *Song et al. (2015)* |
| RA | HAND2-AS1 | MSC | FLS, MH7A | Up-regulated | miR-143-3p/TNFAIP3/NF-κB signaling pathway | *Su et al. (2021)* |
| RA | HOTTIP | FLS | Tcell | Up-regulated | miR-1908-5p/STAT3/ JAK2-STAT3 signaling pathway | *Yao et al. (2021)* |
| OA | LYRM4-AS1 | BMSC | Chondrocyte | Down-regulated | GRPR/miR-6515-5p | *Wang et al. (2021)* |
| OA | NEAT1 | BMSC | Chondrocyte | — | miR-122-5p/SESN2/NRF2 | *Zhang & Jin (2022)* |
| OA | MEG-3 | BMSC | Chondrocyte | Down-regulated | —— | *Jin et al. (2021)* |
| OA | OANCT | Macrophage | Chondrocyte | Up-regulated | PI3K/AKT/mTOR signaling pathway | *Lv et al. (2022)* |
| OA | H19 | FLS | Chondrocyte | Down-regulated | miR-106b-5p/TIMP2 | *Tan, Wang & Yuan (2020)* |
| OA | KLF3-AS1 | MSC | Chondrocyte | Down-regulated | miR-206/GIT1 | *Liu et al. (2018a)* |
| OA | PVT1 | Serum | Chondrocyte | Up-regulated | miR-93-5p/HMGB1/ Toll-like receptor 4/NF-κB signaling pathway | *Meng et al. (2020)* |
| OA | HULC | Chondrocyte | Chondrocyte | Up-regulated | miR-372-3p/GSK signaling pathway | *Song et al. (2017)* |
| OA | 196 differential lncRNAs | Seynovial fluid | — | 52 up-regulated, 144 down-regulated | PI3K/AKT signaling pathway, autophagy pathway | *Wu et al. (2022a)* |
| OA | PCGEM1 | Synovial fluid | — | Up-regulated | —— | *Zhao & Xu (2018)* |
| LN | LINC01015 | Plasma | — | Up-regulated | —— | *Flores-Chova et al. (2023)* |
| LN | LINC01986 | Plasma | — | Up-regulated | —— | *Flores-Chova et al. (2023)* |
| LN | AC087257.1 | Plasma | — | Up-regulated | —— | *Flores-Chova et al. (2023)* |
| LN | AC022596.1 | Plasma | — | Up-regulated | | *Flores-Chova et al. (2023)* |
| PBC/ PSC | H19 | Cholangiocyte | Macrophage | Up-regulated | CCL-2/CCR-2 signaling pathway | *Li et al. (2018)*, *Liu et al. (2019)*, *Li et al. (2020a)* |
| pDM | ENST00000584157.1 | Plasma | HSkMC | Up-regulated | Autophagy pathway | *Li et al. (2022)* |
| pDM | ENST00000523380.1 | Plasma | HSkMC | Down-regulated | Autophagy pathway | *Li et al. (2022)* |
| pDM | ENST00000560054.1 | Plasma | HSkMC | Down-regulated | Autophagy pathway | *Li et al. (2022)* |
| SSc | 192 differential lncRNAs | Plasma | — | 53 up-regulated, 139 down-regulated | Bcell receptor signaling pathway, *etc* | *Sun et al. (2023)* |
| SSc | 281 differential lncRNAs | Neutrophil | — | 119 up-regulated, 162 down-regulated | Wnt, AMPK, IL-23, NOTCH pathways | *Li et al. (2020b)* |

**Note:**
RA, rheumatoid arthritis; OA, osteoarthritis; LN, lupus nephritis; erythematosus; PBC, primary biliary cholangitis; PSC, primary sclerosing cholangitis; pDM, primary dermatomyositis; SSc, systemic sclerosis; FLSs, fibroblast-like synoviocytes; PBMC, peripheral blood mononuclear cell; MH7A, a kind of synovial cell; MSC, mesenchymal stem cell; BMSC, human bone marrow mesenchymal stem cell; HSkMC, human skeletal muscle myoblasts cell.

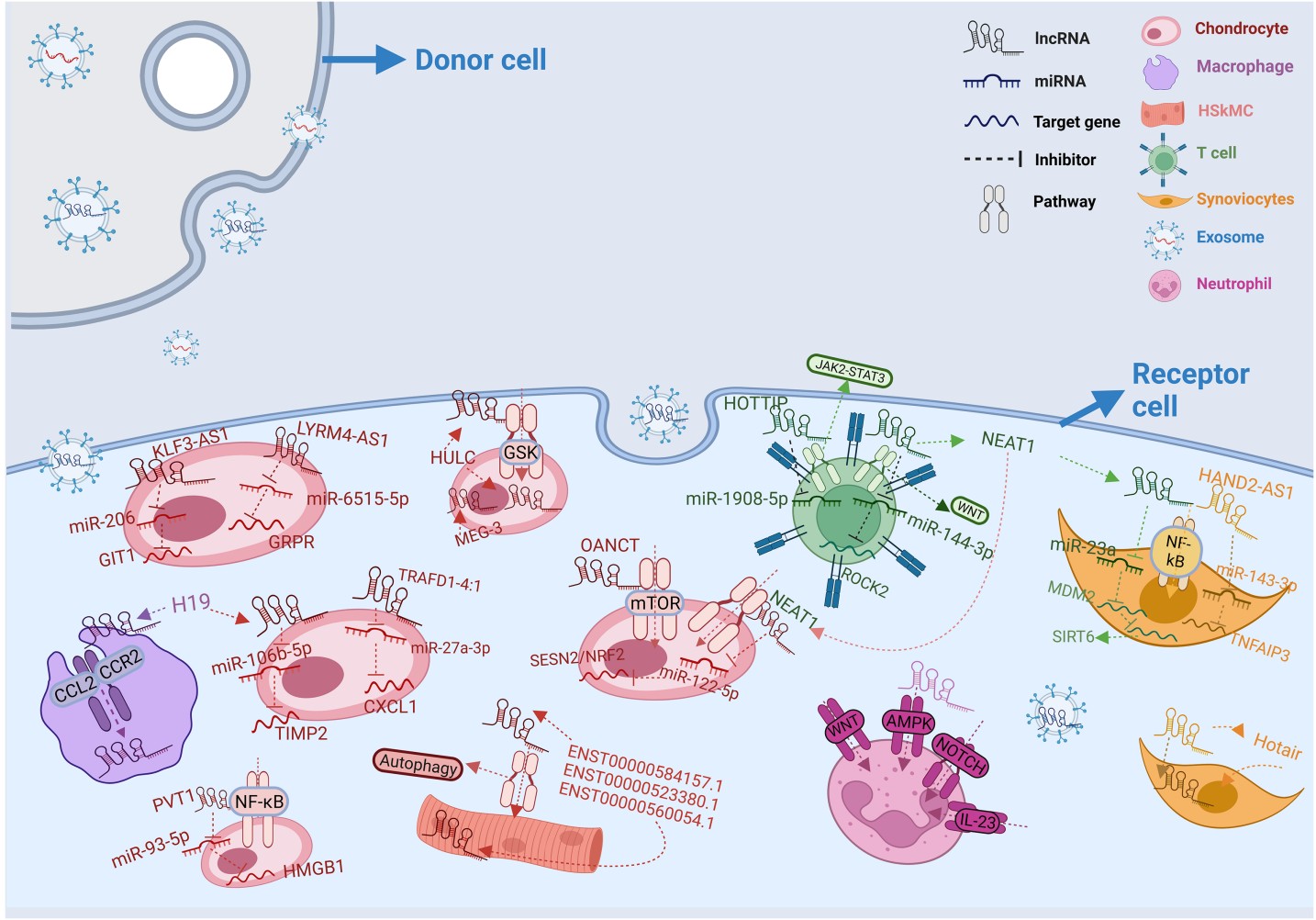

**Figure 1** **Intercellular transfer and function of exosomal lncRNAs.** In rheumatic diseases, exosomes from different donor cells act on different recipient cells by releasing lncRNAs. LncRNAs from donor cells influence mRNA expression in recipient cells by regulating various pathways. In diverse receptor cells, the same lncRNA can play different roles. At the same time, lncRNAs can also cooperatively regulate micRNAs in recipient cells. The legend representation of different types of cells and RNAs is shown in the upper right corner of the figure.

accelerated the degradation of chondrocyte matrix in joints, and inhibited cartilage to play a joint-protective role.

The exosomal lncRNA NEAT1 has been implicated in the abnormal proliferation of FLSs and the inflammation of the synovium by different pathway mechanisms (*Rao et al., 2020*; *Liu et al., 2021*). The NF-κB signaling pathway is known as an important role in the pathogenesis of RA, and p-p65 is a common indicator of the NF-κB signaling pathway activity (*Duan & Li, 2018*). *Rao et al. (2020)* found that the exosomal NEAT1 derived from peripheral blood mononuclear cells (PBMC) is involved in the pathogenesis of RA through the NF-κB signaling pathway. By searching for differential lncRNAs, they found that the NEAT1 expression was significantly higher the healthy control group. Therefore, this lncRNA may be involved in the abnormal activation of FLSs. To further explore the role of NEAT1 in FLSs, they used bioinformatics software to analyze miR-23a and SIRT6.

As downstream genes of NEAT1, these two had a synergistic effect on the pathogenesis of RA through the NF-κB signaling pathway. Meanwhile, they assessed the cell viability of FLSs and the content of inflammatory factors (IL-6, IL-1β, and TNF-α) and p-p65 in serum and found that the levels of the above indexes in the plasma of patients were all increased compared with those of the control group. In addition to human fibroblasts, they injected NEAT1-loaded exosomes in a mice model of RA and found the same result. This further supported their conclusion that the exosomal NEAT1 derived from PBMC may participate in the pathogenesis of RA and could potentially serve as a diagnostic marker in the future.

Exosomal lncRNA Hotair has been implicated in the process of bone destruction in RA (*Song et al., 2015*). Its high expression could be detected in the peripheral blood of patients with RA. Experiments revealed that it could promote the activity of osteoclasts and activate MMP-2 and MMP-13 in synovial cells. The combined effects of Hotair on osteoclast activity and MMP activation synergistically contribute to joint damage and irreversible bone destruction. Thus, Hotair may also become a marker for disease diagnosis in the coming times.

In addition to causing inflammation and bone destruction in diseases, exosomal lncRNAs can exert a protective role. Exosomal lncRNA HAND2-AS1 has been found to inhibit the excessive proliferation of fibroblasts in RA patients through the NF-κB pathway and reduce the release of pro-inflammatory factors IL-6 and TNF-α, thereby preventing the progression of synovitis (*Su et al., 2021*). Further investigation into downstream regulatory genes and target genes revealed that HAND2-AS1 promotes the expression of TNFAIP3 and inhibits the regulation of miR-143-3p leading to the excessive proliferation and migration of FLSs through the miR-143-3p/TNFAIP3 axis. Moreover, HAND2-AS1 promotes the apoptosis of hyperproliferative cells, which reduces the release of inflammatory factors and allows the synovial environment to regain homeostasis. These findings open new possibilities for targeted management of the disease, offering potential therapeutic avenues for treatment.

The competitive endogenous RNA (ceRNA) network serves as a connection between coding RNA and non-coding RNA (such as microRNA, lncRNA, and siRNA, *etc.*) and this network enables the interaction and communication between these RNAs, leading to the enrichment of biological pathways, providing a possibility to explore the biological signaling pathways involving these genes (*Qi et al., 2015*). To note, it is common that a single gene in a disease can be involved in different signaling pathways. Through bioinformatics analysis, it has been discovered that the collaboration of these pathways often contributes to the occurrence of diseases. This coordination among pathways contributes to the complexity of disease pathogenesis. NEAT1, a special non-coding RNA, not only plays a role in the enrichment of the NF-κB signaling pathway, but also affects the proliferation and differentiation of T cells through the Wnt/β-catenin signaling pathway, contributing to the RA pathogenesis.

The Wnt/β-catenin signaling pathway plays an crucial role in various processes, such as kidney injury and repair and tumor progression (*Schunk et al., 2021*; *Bian et al., 2020*). More recently, IL-35, an anti-inflammatory cytokine from the IL-12 cytokine family, has

been found to stimulate the differentiation of osteoblasts through the Wnt/β-catenin signaling pathway and inhibit bone loss in patients with RA (*Li et al., 2019*). This suggests that genes enriched in the Wnt/β-catenin signaling pathway may also be involved in the pathogenesis of RA. Based on this point, *Liu et al. (2021)* found that NEAT1 promotes the proliferation, hyper-differentiation, and migration of T17 cells by the Wnt signaling pathway, causing the imbalance of Th17/Treg cells. This imbalance has a profound impact on the pathogenesis of RA.

The imbalance of CD4+ T lymphocyte subsets plays a key role in RA. Th17 cells promote inflammation, while Treg cells counteract the effects of Th17 cells, maintaining immune system homeostasis (*Niu et al., 2012*). *Yao et al. (2021)* conducted a study using a self-built RA mouse model and observed the number of exosomal LncRNA HOTTIP derived from synovial fibroblasts (RASFs) was higher than that in normal mice. They also noted differences in the number of T cells in the spleen tissues of the these mice. To determine which pathway is responsible for HOTTIP production and how the number of T cells is changed, they examined the downstream genes miR-1908-5p and STAT3 and found that these genes were jointly enriched in the JAK2-STAT3 signaling pathway. *In vitro* experiments involving the addition of HOTTIP to the CD4+ T cell population demonstrated a significant increase in Th17 cells a concomitant decrease in Treg cell proportions. This imbalance of Th17/Treg cells stimulated the production of a large number of pro-inflammatory factors, leading to synovial tissue damage. Based on these findings, HOTTIP plays a harmful role in RA, and inhibiting its regulation of CD4+ T lymphocyte subsets can prevent the aggravation of joint damage.

In summary, the comprehensive investigations discussed in this review have substantiated the significance of exosomal lncRNAs in the occurrence of RA. The stable expression of exosomes in the blood increases the diversity of laboratory diagnoses of the disease, and genes with protective effects bring new hope to the patients.

## Osteoarthritis (OA) and exosomal lncRNA

OA is another common disease in rheumatology. Compared with RA, OA is not an autoimmune disease, but a degenerative disease related to age, obesity, injury, and congenital deformity (*Grässel & Muschter, 2020*). Clinically, OA often presents as pain and injury in large joints and weight-bearing joints. It can also affect small joints, particularly the distal interphalangeal joints, which distinguishes it from RA. The damage of chondrocytes, the formation of osteophytes, and the imbalance between extracellular matrix synthesis and catabolism are key factors contributing to progressive destruction of articular cartilage and cartilage loss, which are considered as the initial conditions and progression factors leading to the occurrence of OA (*Guilak et al., 2018*; *Fellows, Matta & Mobasheri, 2016*). Currently, the diagnosis of OA remains exclusive, lacking a universally recognized specific antibody or targeted drug for treatment. By reviewing the literatures in recent years, it is found that some studies have been committed to finding breakthroughs in the treatment and diagnosis of OA from exosome lncRNAs. Moreover, marrow mesenchymal stem cells (BMSCs), chondrocytes, macrophages, synovial cells were associated with these exosomal lncRNAs.

As multifunctional cells, BMSCs possess the remarkable ability to self-replicate and differentiate into various cell lineages. The differentiated cells have the capacity to regulate the immune system and counteract non-specific inflammation, making them valuable in the treatment of various diseases (Chu et al., 2020; Neri, 2019). Exosomes derived from BMSCs can reduce the damage of chondrocytes and inhibit the formation of new osteophytes by regulating the phenotypic transformation of macrophages, preventing the onset of OA (Zhang et al., 2020). Furthermore, certain exosomal lncRNAs have exhibited regulatory effects on chondrocytes.

In an *in vitro* chondrocyte injury model, Wang et al. (2021) investigated the regulatory role of exosomal LncRNA LYRM4-AS1 in IL-1β-induced chondrocyte apoptosis through the LYRM4-AS1/miR-6515-5p/GRPR axis. Subsequently, they then injected the exosomes into the model mice and found that the content of extracellular matrix protein and its effect on protecting chondrocytes were increased compared with those before injection, while the content of MMP-13 and its effect on degrading cartilage was decreased. Therefore, the exosomes derived from BMSCs had a certain protective effect on chondrocytes. Moreover, they performed a qRT-PCR analysis in exosome-injected chondrocytes and determined that LYRM4-AS1 counteracted the detrimental effects of IL-1β on chondrocytes *in vitro*. The therapeutic impact of GRPR was observed in reversing the effects of LYRM4-AS1,, and miR-6515-5p influenced the beneficial effect of LYRM4-AS1.

Zhang & Jin (2022) found that exosomal LncRNA NEAT1, also from BMSCs, acted on chondrocytes through exosome transport and inhibited the expression of miR-122-5p. Moreover, NEAT1 protects joints by inhibiting chondrocyte apoptosis in OA by regulating the miR-122-5p/SESN2/NRF2 axis *in vivo*.

In another study, exosomal lncRNA MEG-3 was shown to treat mouse chondrocytes and inhibit apoptosis (Jin et al., 2021). The researchers knocked out MEG-3 in chondrocytes induced by IL-1β and found that the intensity of β-gal staining used to evaluate chondrocyte senescence was decreased. In addition, the number of chondrocytes in the culture medium increased compared to the before apoptosis state. Therefore, MEG-3 can regulate the apoptosis of chondrocytes, and providing strong evidence for its potential in the treatment of OA by stem cells.

Among numerous inflammatory diseases, M1 type macrophage is a significant class of recognized cells with functions involved in multiple inflammatory pathways. Lv et al. (2022) identified a previously undiscovered lncRNA with high expression through high-throughput sequencing and named it OANCT which enriched in PI3K/AKT/mTOR pathway. The expression level of OANCT was positively correlated with K-L stage in OA patients. Moreover, OANCT can be secreted by damaged chondrocytes in OA model rats and further aggravate joint inflammation by promoting macrophages to M1 type polarization.

Apart from chondrocytes, synovial cells are also involved in OA (Pan et al., 2017). In their research on the relationship between synovitis and chondrocytes, Tan, Wang & Yuan (2020) found that the lncRNA H19 in exosomes derived from FLSs protected damaged chondrocytes from apoptosis. In addition, overexpressed H19 exosomes reversed

IL-1β-induced damage to chondrocytes, offering another possibility for the future OA treatment.

Besides the above exosomal lncRNAs, KLF3-AS1 derived from BMSCs has been shown to promote chondrocyte proliferation and inhibit apoptosis through the KLF3-AS1/miR-206/GIT1 axis in both human and mouse chondrocytes and further suppresses the destruction of cartilage (*Liu et al., 2018a*). PVT1 could act on miR-93-5p through sponge action and regulate the HMGB1/Toll-like receptor 4/NF-κB pathway, inhibiting the expression of PVT1 and alleviating the progression of OA (*Meng et al., 2020*). Through high-throughput sequencing, nine exosomal lncRNAs identified to exhibit differences between patients with osteoarthritis and healthy individuals. Among them, HULC has been confirmed to contribute to chondrocyte apoptosis and participate in the GSK signaling pathway, which may be closely related to the pathogenesis of OA (*Song et al., 2017*). Another high-throughput sequencing detected differential ncRNAs in articular fluid, including 196 lncRNAs, which were found to be enriched in PI3K/Akt and autophagy pathway through analysis of ceRNA network (*Wu et al., 2022a*).

Other than that, another study compared the special exosomes in synovitis in patients with early and advanced OA through differential analysis, qRT-PCR validation, and ROC curve and found that lncRNA PCGEM1 may become the basis for early diagnosis (*Zhao & Xu, 2018*). Taken together, all these findings on lncRNAs provide a new cornerstone for the specific diagnosis of OA in the future and enrich the existing treatment methods.

## Systemic lupus erythematosus (SLE) and exosomal lncRNA

SLE is a disease involving multiple organs and systems of the whole body. Its pathological basis is mainly vasculitis in the affected tissues, accompanied by abnormal inflammatory cell infiltration, immune complex deposition in the affected organs, *etc.* The pathogenesis is related to sex hormones, epigenetics, environmental factors, infection and many other factors (*Tian et al., 2023*).

According to the latest epidemiological statistics in 2023, the global incidence of this disease is about 5.14 per 100,000 people, and the number of new cases can reach 400,000 people every year (*Zen et al., 2023*). The harm caused by infection and irreversible damage of important organs leads to about 18.6/1,000 deaths per year (*Moghaddam et al., 2021*), among which kidney involvement ranks among the top three causes of death in many studies (*Dhital et al., 2020*). Clinically, lupus nephritis can be manifested in various manifestations such as hematuria, proteinuria, cellular uria, and renal failure (*Gasparotto et al., 2020*), which can be clearly diagnosed by pathological type of kidney biopsy. At present, some studies have reported specific indicators of lupus nephritis (*Chen et al., 2018*; *Koutsonikoli et al., 2017*), but they have not been verified by cross-regional populations and large samples.

*Flores-Chova et al. (2023)* analyzed by GO annotation and KEGG pathways multiple ncRNAs involved in inflammation, fibrosis and other pathways in lupus nephritis, including four up-regulated exosomal lncRNAs (LINC01015, LINC01986, AC087257.1, AC022596.1). By constructing lncRNA-miRNA networks between these four lncRNAs and other miRNA, they searched for the target genes. Together, these target genes were

involved in inflammatory pathways such as p53, MAPK, and Ras pathway, other signaling pathways that regulate cell aging, stem cell pluripotency, thereby identifying a number of potential candidate genes as targets with therapeutic significance.

## Autoimmune liver diseases and exosomal lncRNA

Autoimmune liver diseases are a group of autoimmune diseases involving the liver, including autoimmune hepatitis (AIH), primary biliary cholangitis (PBC), primary sclerosing cholangitis (PSC), and overlap syndrome (OS). PBC and PSC are cholestatic autoimmune diseases. In addition to liver pathology, the detection of serological specific antibodies represented by antimitochondrial antibodies serves as an important diagnosis criterion for these diseases. In cases where individuals exhibit high suspicion of these diseases but test negative for the aforementioned antibodies, liver pathological examinations remain essential. Unfortunately, the worldwide adoption of liver biopsy and histopathology is limited due to variations in medical staff expertise and patient preferences.

In cholestatic liver disease, the accumulation of excessive of endogenous bile acids damages cholangiocytes, disruption of bile acid transporters and nuclear receptors, and changes the composition of intestinal microbiota due to bile duct structure abnormalities or abnormal bile secretion (*Zhou et al., 2023*). These observations indicate that, cholangiocytes become the main target cells. With the widespread application of high-throughput sequencing in experiments, many studies have identified the involvement of exosome-mediated non-coding RNAs in the pathogenesis of diseases. However, only a limited number of studies have been conducted in relation to cholestatic liver disease. To date, research on exosomal lncRNA H19 has been relatively comprehensive and may hold potential as a breakthrough point for future diagnosis and treatment strategies.

*Li et al. (2018)* conducted a study in which they examined cholangiocytes from patients with PBC, patients with PSC, and mice. They discovered that the exosomes derived from cholangiocytes inhibited the expression of small heterodimeric chaperone receptor (SHP) in hepatocytes by transporting H19, thus affecting the secretion of cholesterol and bile acids. They detected the level of exosomal H19 in the serum of patients with compensated PSC and found significantly higher expression of H19 in patients with liver cirrhosis compared to healthy individuals. In a mouse model of liver cirrhosis, they found that the injection of H19 derived from severe liver damage into mildly damaged liver promoted the progression of liver fibrosis. Interestingly, injecting the same exosomes into normal mice did not cause liver damage, suggesting that H19 is not the initiating cause of cholestatic liver damage but drives the progression of the entire disease.

Previous studies have revealed that hepatic stellate cells (HSCs) are essentially macrophages and are key cells in liver fibrosis (*Higashi, Friedman & Hoshida, 2017*). To explore whether cholangiocyte-derived lncRNA-H19 regulates the activation of HSCs, *Liu et al. (2019)* found that exosomes derived from cholangiocytes were preferentially absorbed by HSCs in many liver cells, indicating that exosomes can establish a communication pathway between stellate cells. By culturing primary HSCs, they found that the H19 carried by exosomes promoted the proliferation and differentiation of

primary cells, leading to an up-regulation in the expression of fibrosis marker genes. This finding proved that H19 affects HSCs and promotes liver fibrosis by activating the proliferation and differentiation processes.

As a lncRNA, H19 exerts its effects on HSCs through a certain pathway. *Li et al.*'s *(2020a)* continued to make a thorough inquiry into this issue and found that H19 promoted macrophage activation through the CCL-2/CCR-2 signaling pathway, resulting in cholestatic liver injury. In the liver, the excessive accumulation of bile acids can trigger the rapid release of various inflammatory factors, including a specific chemokine—CCL-2. Its increase might recruit some pro-inflammatory monocytes to the site of injury, thus forming a local inflammatory response and resulting in bile duct epithelial cell damage (*Baeck et al., 2012*). Based on this, they found that the exosomal lncRNA H19 derived from cholangiocytes stimulated the differentiation of macrophages and accelerated the differentiation through positive feedback by promoting the release of cytokines, such as CCL-2, TNF-$\alpha$, and IL-6. The presence of CCL-2 was significantly low in H19-free mouse macrophages, thereby highlighting its importance in the overall process. To support the results of the *in vitro* experiments, they knocked out H19 and found the significantly down-regulated CCR-2 expression and hindered macrophage differentiation in Mdr2-/- and BDL cholestasis mouse models. This finding proved that exosomes carrying H19 could promote liver fibrosis and thus can be used as a new diagnostic marker. When used as a therapeutic target, this gene might delay the progression of liver fibrosis in patients with PBC and PSC, thereby prolonging patient survival and providing support for liver transplantation.

## Primary dermatomyositis (DM) and exosomal lncRNA

DM, which is mainly classified as either primary or secondary, is a group of idiopathic inflammatory myopathies characterized by characteristic skin lesions and heterogeneous systemic multisystem damage (*DeWane, Waldman & Lu, 2020*). The pathological features of the skin can be hyperkeratosis, epidermal atrophy, vacuolar interface dermatitis, basement membrane thickening, and endothelial cell injury around blood vessels composed of CD4+ lymphocytes. The pathological features of muscle may be perifascial tissue atrophy, capillary complement deposition in muscle intima, and infiltration of perimembranous or perivascular inflammatory cells (*Tanboon & Nishino, 2019*). Secondary DM is often closely related to malignant tumors, whereas primary DM is an autoimmune disease that is relatively easy to diagnose but difficult to treat in many rheumatic diseases. With the emergence of the myositis-related antibody spectrum, step-by-step progress has been made in the subtype diagnosis of DM. At present, glucocorticoids and traditional immunosuppressants are still the main treatment regimens for dermatomyositis. Although some case reports and small randomized controlled trials (RCT) have proven that some cytokine inhibitors, targeted drugs acting on lymphocytes, and drugs related to intercellular signaling pathway inhibitors have shown varying effects in improving skin changes in DM and interstitial lung lesions, such claims lack substantial clinical data (*Kodumudi et al., 2022*). Given the utility of lncRNAs in the diagnosis and treatment of other rheumatic diseases, the value of lncRNAs in the diagnosis of DM is

gradually being discovered. Some research teams are optimistic about utilizing these exosomes at the diagnosis and treatment, aiming to achieve significant advancements in the field.

Autophagy is a normal physiological process in which cells transfer cytoplasmic contents to lysosomes through various pathways for degradation (*Mizushima et al., 2008*). Although autophagy facilitates cellular recycling and energy production, it is essentially a pathway of cellular destruction. In previous research, it has been found that autophagy is closely related to neurodegenerative diseases, α1-antitrypsin deficiency, and some special hereditary cardiomyopathies (*Martinez-Vicente & Cuervo, 2007*; *Perlmutter, 2006*; *Terman & Brunk, 2005*). More recently, excessive activation of autophagy has been found to disrupt skeletal muscle cells, leading to inflammatory myopathy (*Cappelletti et al., 2014*). In addition, the genes involved in the autophagy pathway show potential in diagnosing dermatomyositis (*Wang, Fang & Liu, 2022*). Considering this, it raises the question of whether exosomal lncRNAs associated with autophagy also possess value in this context.

In this context, *Li et al. (2022)* found 452 differentially expressed lncRNAs by measuring the whole transcriptome of peripheral blood exosomes in a small sample of patients with DM. To investigate whether these peripheral blood lncRNAs are involved in muscle injury, they compared these genes with a lncRNA gene pool related to muscle injury and found nine overlapping genes. Among these, three lncRNAs (ENST00000584157.1, ENST00000523380.1, and ENST00000560054.1) performed similar expression characteristics, as shown in Fig. 1 and Table 2. Further analysis through KEGG pathway analysis revealed that these genes and their target genes were co-enriched in the autophagy pathway. Finally, they verified these results by qRT-PCR, which revealed upgraded expression of lncRNA ENST00000584157.1 and downgraded expression of lncRNAs ENST00000523380.1 and ENST00000560054.1, suggesting these three special lncRNAs could regulate the autophagy of skeletal muscle myoblasts in the plasma exosomes of patients with DM and contribute to the procession of muscle injury. Overall, these findings pave the way for potential future investigations into exosomal lncRNAs in DM, offering hope for improved diagnosis and treatment of this disease.

## Systemic sclerosis (SSc) and exosomal lncRNA

SSc, also known as scleroderma, is a rheumatic disease with unknown etiology and high mortality, characterized by fibrosis of the skin and internal organs, as well as vascular lesions (*Denton & Khanna, 2017*). Pathological manifestations of microvascular disease and endothelial cell dysfunction, under the influence of inflammatory cells and cytokines, a large number of endothelial cells into myofibroblasts, resulting in different degrees of tissue and organ damage and fibrosis (*Cutolo, Soldano & Smith, 2019*). Some patients present with specific skin-related clinical manifestations before experiencing involvement of internal, and in such cases, a diagnosis can be made by skin biopsy. However, many patients experience internal organ involvement as the first symptom, and most of them have poor prognoses and treatment effects. Therefore, the early clinical detection of skin changes, specific antibodies, and gene tests have become the key to diagnosing this disease.
Complex autoimmune responses, including innate and adaptive immunity and the production of specific autoantibodies, are the basis of systemic sclerosis (*Thoreau et al., 2021*). As an important link to innate immunity, neutrophils play a dual role in rheumatic diseases. On one hand, they possess a protective function through phagocytosis, helping to clear pathogens and debris. On the other hand, abnormal activation of neutrophils can lead to the release of reactive oxygen species (ROS) and proteases outside the cell, causing tissue damage and disrupting oxidative homeostasis (*El-Benna et al., 2016*). So far, neutrophils are involved in the occurrence of a variety of rheumatic diseases. For example, in ANCA-associated vasculitis, the excessive activation of neutrophils induces neutrophil extracellular traps (NETs) formation, which is involved in the formation of ANCAs and the damage of systemic small blood vessels (*Söderberg & Segelmark, 2016*). Neutrophils are also associated with endothelial damage in systemic sclerosis-related blood vessels, which can lead to vascular remodeling and inflammation (*Arad et al., 2011*).

*Sun et al. (2023)* constructed the lncRNA-miRNA-mRNA network interaction map of exosomes in systemic sclerosis, in which a total of 192 lncRNAs with differential expression were detected. By double luciferase reporter gene detection, it was found that lncRNA ENST00000313807 interacted with miR-29a-3p, meanwhile miR-29a-3p interacted with COL1A1. Correlation analysis showed that these three genes were correlated with imaging results of systemic sclerosis lung interstitial disease, C-reactive protein level, SCL-70 antibody titer and other clinical indicators. ROC curve analysis showed that these genes were all significant in the diagnosis of SSc.

*Li et al. (2020b)* studied the miRNA and lncRNA profiles of neutrophil-derived exosomes in the plasma of patients with diffuse scleroderma. From the genealogy, they found 22 miRNAs and 281 lncRNAs with differential expressions that were enriched in Wnt, AMPK, IL-23, and NOTCH signaling pathways. In the future, these special genes and pathways may become potential diagnostic markers and therapeutic targets to enrich the existing diagnosis and treatment strategies.

## CONCLUSION

In this review, we summarized the research results of exosomal lncRNAs in some common rheumatic diseases. In addition, we also tried to summarize the research results on exosomal lncRNAs in other rheumatic diseases, such as polymyalgia rheumatica and ankylosing spondylitis. So far, there is no research in these fields. However, lncRNA has been found in these diseases; for example, lncRNA-NEF in ankylosing spondylitis has a role in evaluating treatment effect and predicting disease recurrence (*Han et al., 2022*). We will continue to follow new research development.

Although the research on exosomal lncRNAs have not been translated into results in the field of rheumatic diseases, the future is promising. In various studies on exosomal lncRNAs in rheumatic diseases, we can find various links in which exosomal lncRNAs with differential expression can participate. These new findings bring us to think that by detecting the levels of genes that are highly expressed in the body, such as the exosomal Hotair and NEAT1 in rheumatoid arthritis, it may be possible to help clinicians better diagnose the disease. Some research, such as exosomal PCGEM1, may fill the gap in the

early diagnosis of rheumatic diseases. In addition, through in-depth study of the target genes and common pathways involved in the action of these genes, it was found that the occurrence of rheumatic diseases was caused by the abnormal activation of various immune pathways and the release of various inflammatory factors, which also provided a theoretical basis for our better understanding of diseases and targeted therapy of these pathways and inflammatory factors.

In addition to having diagnostic clinical significance, the differentially expressed low-expression genes can be discovered and used by more graduate students in the treatment of rheumatic diseases now or in the future to break the existing traditional treatment model. The addition of exosomes provides a good carrier for this treatment. After all, it has a recognized high safety and high stability. In a study on the treatment of bone tumors, MEG3 coated with modified exosomes can be efficiently diverted to tumor cells *in vivo* and *in vitro* for the treatment of bone tumors (*Huang et al., 2022*). In rheumatic diseases, we also hope that exosome lncRNAs can be used to treat rheumatic diseases more accurately in actual clinical practice by modifying the transport of exosomes or special materials.

### Funding
The authors received no funding for this work.

### Competing Interests
The authors declare that they have no competing interests.

### Author Contributions
- Ruofei Chen conceived and designed the experiments, analyzed the data, prepared figures and/or tables, and approved the final draft.
- Dongqing Zhou performed the experiments, analyzed the data, prepared figures and/or tables, and approved the final draft.
- Yangfan Chen performed the experiments, analyzed the data, prepared figures and/or tables, and approved the final draft.
- Mingwei Chen conceived and designed the experiments, authored or reviewed drafts of the article, and approved the final draft.
- Zongwen Shuai conceived and designed the experiments, authored or reviewed drafts of the article, and approved the final draft.

### Data Availability
This is a literature review.

### Supplemental Information
Supplemental information for this article can be found online at http://dx.doi.org/10.7717/peerj.16434#supplemental-information.

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
