# Peer review of "Understanding the role of exosomal lncRNAs in rheumatic diseases: a review"

_PeerJ, doi:10.7717/peerj.16434_

## Round 0.1 · original submission · Major Revisions

Authors should revise according to the suggestions of reviewers. The modifications should be marked. A point to point response letter is needed.

**Language Note:** The review process has identified that the English language must be improved. PeerJ can provide language editing services - please contact us at [email protected] for pricing (be sure to provide your manuscript number and title). Alternatively, you should make your own arrangements to improve the language quality and provide details in your response letter. – PeerJ Staff

Reviewer 1 ·

Basic reporting

The article discusses the role of exosomal long non-coding RNAs (lncRNAs) in rheumatic diseases, including rheumatoid arthritis (RA), osteoarthritis (OA), systemic lupus erythematosus (SLE), autoimmune liver diseases, primary dermatomyositis, and systemic sclerosis. Exosomes, small vesicles secreted by cells, play a role in various physiological and pathological processes by delivering biologically active substances such as nucleic acids, proteins, and lipids to recipient cells.

Exosomal lncRNA NEAT1 derived from peripheral blood mononuclear cells (PBMC) is involved in the pathogenesis of RA through the NF-κB signaling pathway. Exosomal lncRNA Hotair contributes to bone destruction in RA by promoting osteoclast activity. On the other hand, exosomal lncRNA HAND2-AS1 inhibits fibroblast proliferation and reduces the release of pro-inflammatory factors, potentially preventing synovitis progression in RA. Exosomal lncRNA LYRM4-AS1 regulates chondrocyte apoptosis, while NEAT1 inhibits chondrocyte apoptosis and protects joints from damage. Other exosomal lncRNAs, such as MEG-3, KLF3-AS1, PVT1, and HULC, have also been implicated in the pathogenesis of OA. These lncRNAs are involved in inflammatory pathways and may serve as potential diagnostic markers and therapeutic targets. Autoimmune liver diseases, including autoimmune hepatitis, primary biliary cholangitis, primary sclerosing cholangitis, and overlap syndrome, are characterized by liver inflammation and damage. Exosomal lncRNA H19 derived from cholangiocytes has been found to promote liver fibrosis by activating hepatic stellate cells (HSCs) and promoting macrophage activation through the CCL-2/CCR-2 signaling pathway. Exosomal lncRNAs associated with autophagy have been implicated in muscle injury in DM, suggesting their potential diagnostic value. Systemic sclerosis connective tissue disease characterized by fibrosis and vascular lesions. Exosomal lncRNA ENST00000313807 has been found to interact with miR-29a-3p and COL1A1, which are associated with systemic sclerosis lung interstitial disease and other clinical indicators.

Experimental design

The survey methodology for this study involved conducting a systematic literature review using various scientific databases. The databases searched included PubMed, Web of Science, Embase, and Cochrane. The search strategy involved using specific keywords and combinations of keywords related to exosomes, lncRNAs, and different rheumatic diseases such as rheumatoid arthritis, osteoarthritis, systemic lupus erythematosus, primary biliary cholangitis, dermatomyositis, and systemic sclerosis. By conducting a comprehensive search using these databases and search terms, the researchers aimed to gather relevant articles and studies that explored the role of exosomal lncRNAs in rheumatic diseases. This systematic approach helps ensure that a wide range of relevant literature is considered for the review, enhancing the reliability and comprehensiveness of the findings.

Validity of the findings

No comments

Additional comments

1. Provide more specific information about the search strategy and inclusion/exclusion criteria used to select the studies. This would enhance transparency and allow readers to assess the comprehensiveness and potential biases of the review.\
2. Discuss the potential practical implications of the findings for clinicians, researchers, or policymakers. Identify any potential applications of the reviewed evidence in clinical practice, patient management, or future research directions. This would help bridge the gap between research and real-world impact.
3. Figures presented have bit lower quality and has been generated using trail version of Biorender, if possible try to get good quality image and detail briefing with the figure legends.
4. Rephrase the conclusion as "Emphasize the need for standardized methods for exosome extraction that ensure high purity and reproducibility. Highlight the ongoing efforts in the field to develop standardized protocols and techniques that can be easily implemented in clinical settings. Discuss the potential impact of standardized extraction methods on the widespread application of exosomal.lncRNAs in clinical practice."
5. If possible add "complexity of exosomal lncRNAs and their diverse roles in different pathways and cellular processes. Stress the importance of further research to elucidate the underlying mechanisms by which exosomal lncRNAs contribute to the pathogenesis and progression of rheumatic diseases. This understanding is crucial for identifying therapeutic targets and developing targeted interventions."

Reviewer 2 ·

Basic reporting

This manuscript does not fulfi the standards established for the journal to be considered for publication.

Experimental design

As a review, it is only a tautology of words, and summary words and figures are not enough.

Validity of the findings

no comment

Additional comments

The language needs further modification by native speakers.

Reviewer 3 ·

Basic reporting

The authors of this manuscript review the impact of exosomal lncRNAs in various rheumatic diseases like rheumatoid arthritis, osteoarthritis, lupus, and more, offering potential insights into disease mechanisms and therapeutic approaches. Understanding these lncRNAs and associated signaling pathways could provide new avenues for managing rheumatic diseases. Overall, content of this manuscript is valuable. However, it needs some minor revisions.

1- For the whole article, there are grammatical errors and issues with sentence formation. The authors need to review the manuscript.
2- Can authors add Figure description for Figure 1?
3- There are few other rheumatoid diseases as well. eg. Polymyalgia Rheumatica (PMR) etc. Did authors want to focus on some of them? If yes, may be mention in the introductory text.
4- In the section - Rheumatic diseases and exosomal lncRNAs, authors can introduce these diseases in detail and talk about pathology before mentioning the role of exosomes.

Experimental design

NA

Validity of the findings

NA

Additional comments

NA

---

## Round 0.2 · accepted · Accept

Despite not being able to get a response from 2 reviewers, another reviewer has approved the revised paper for publication. I was satisfied with the responses and revisions made by the authors. The Reviewer's concerns have been well addressed. I believe that this revised manuscript is ready to be considered for publication in this journal.

The title is ungrammatical - we suggest changing it to:

"Understanding the role of exosomal lncRNAs in rheumatic diseases: a review"

Reviewer 3 ·

Basic reporting

NA

Experimental design

NA

Validity of the findings

NA